# Detecting, Explaining, and Mitigating Memorization in Diffusion Models

**Yuxin Wen**[1]*, **Yuchen Liu**[2]*, **Chen Chen**[3], **Lingjuan Lyu**[3]†
[1]University of Maryland, [2]Zhejiang University, [3]Sony AI
`ywen@umd.edu, yuchen.liu.a@zju.edu.cn`
`{ChenA.Chen,Lingjuan.Lv}@sony.com`

## Abstract

Recent breakthroughs in diffusion models have exhibited exceptional image-generation capabilities. However, studies show that some outputs are merely replications of training data. Such replications present potential legal challenges for model owners, especially when the generated content contains proprietary information. In this work, we introduce a straightforward yet effective method for detecting memorized prompts by inspecting the magnitude of text-conditional predictions. Our proposed method seamlessly integrates without disrupting sampling algorithms, and delivers high accuracy even at the first generation step, with a single generation per prompt. Building on our detection strategy, we unveil an explainable approach that shows the contribution of individual words or tokens to memorization. This offers an interactive medium for users to adjust their prompts. Moreover, we propose two strategies i.e., to mitigate memorization by leveraging the magnitude of text-conditional predictions, either through minimization during inference or filtering during training. These proposed strategies effectively counteract memorization while maintaining high-generation quality. Code is available at https://github.com/YuxinWenRick/diffusion_memorization.

## 1 Introduction

Recent advancements in diffusion models have revolutionized image generation, with modern text-to-image diffusion models, such as Stable Diffusion and *Midjourney*, demonstrating unprecedented capabilities in generating diverse, stylistically varied images. However, a growing body of research (Somepalli et al., 2022; Carlini et al., 2023; Somepalli et al., 2023b) reveals a concerning trend: some of these "novel" creations are, in fact, near-exact reproductions of images from their training datasets, as depicted in the top row of Fig. 1. Some of the creations appear to borrow elements from these training images, as illustrated in the second row of Fig. 1. This issue of unintended memorization poses a serious concern for model owners and users, especially when the training data contains sensitive or copyrighted material. A poignant real-life example of this is *Midjourney*, which felt obliged to ban prompts with the substring "Afghan" to avoid generating images reminiscent of the renowned copyrighted photograph of the Afghan girl. Yet, as Wen et al. (2023a) note, merely banning the term "Afghan" does not prevent the model from recreating those images. In light of these issues, the development of techniques to detect and address such inadvertent memorizations has become crucial.

To address this, we first introduce a novel method for detecting memorized prompts. We've observed that for such prompts, the text condition consistently guides the generation towards the memorized solution, regardless of the initializations. This phenomenon suggests significant text guidance during the denoising process. As a result, our detection method prioritizes the magnitude of text-conditional predictions as its cardinal metric. Distinctively, memorized prompts tend to exhibit larger magnitudes than non-memorized ones, as showcased in Fig. 1. Unlike previous methods that query large training datasets with generated images (Somepalli et al., 2022) or assess density across multiple

---

*Work done during an internship at Sony AI.

†Corresponding author.

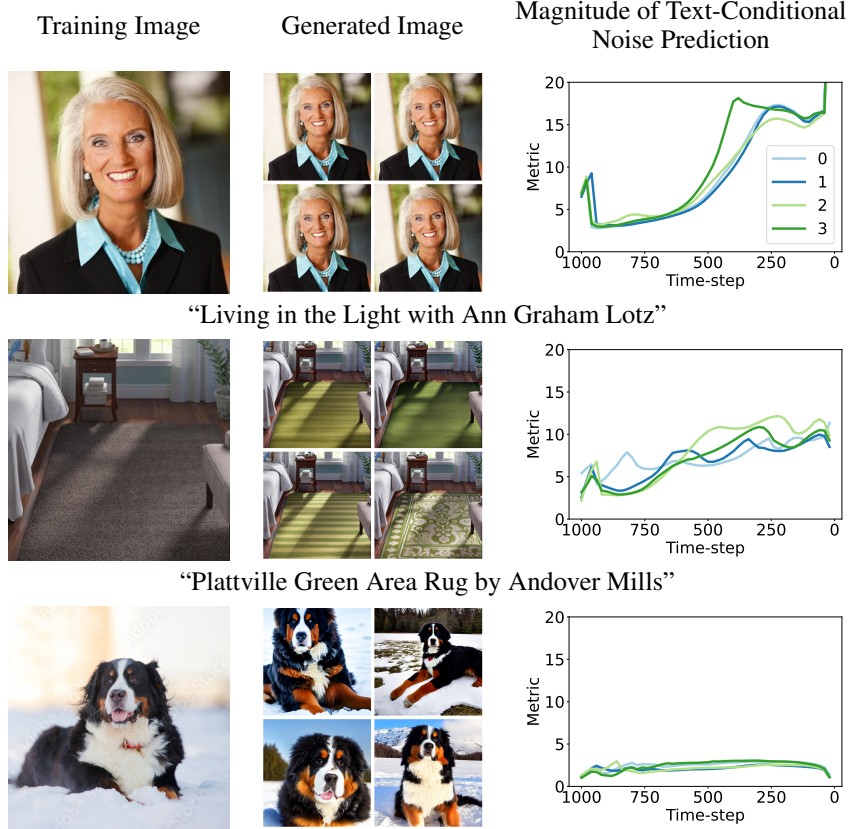

Figure 1: Memorization vs. non-memorization generation. We display the magnitude of text-conditional noise prediction at each time-step, as described in Section 3.3, for all four generations distinctly (with 4 different random seeds) for each prompt. As illustrated in the first two rows, the metric typically indicates a higher value when memorization occurs. On the other hand, the normal generations, represented in the third row, consistently exhibit significantly lower metric values.

generations (Carlini et al., 2023), our strategy offers precise detection without adding any extra work to the existing generation framework and even without requiring multiple generations. In terms of efficacy, our method achieves an AUC of 0.960 and a TPR@1%FPR of 0.760 in under 2 seconds. In contrast, the baseline method (Carlini et al., 2023) demands over 39 seconds, even though it registers an AUC of 0.934 and a TPR@1%FPR of 0.523. Such efficiency empowers model owners to halt generations early and initiate corrective measures promptly upon detecting a memorized prompt.

Building on our discoveries, we devise a strategy to highlight the influence of each token in driving memorization, aiming to pinpoint the specific trigger tokens responsible for it. Following the intuition that removing these trigger tokens should neutralize memorization, we anticipate a corresponding reduction in the magnitudes of text-conditional predictions. Thus, by evaluating the gradient change for every token when minimizing text-conditional prediction magnitudes, we discern the relative significance of every token to memorization. Armed with this tool, model owners can provide constructive feedback to users, guiding them to identify, modify, or omit these pivotal trigger tokens, effectively reducing the propensity for memorization. In contrast to earlier work like (Somepalli et al., 2023b), where trigger tokens were discerned manually from training data or by experimenting with various token combinations, our approach stands out for its automated nature and computational efficiency.

Lastly, we introduce mitigation strategies to address memorization concerns. Catering to both inference and training phases, we present model owners with a choice of two distinct tactics. For inference, we suggest using a perturbed prompt embedding, achieved by minimizing the magnitude of text-conditional predictions. During training, potential memorized image-text pairs can be screened out based on the magnitude of text-conditional predictions. Our straightforward approaches

ensure a more consistent alignment between prompts and generations, and they effectively reduce the memorization effect when benchmarked against baseline mitigation strategies.

## 2 RELATED WORK

**Membership Inference.** The membership inference attack (Shokri et al., 2017) aims to determine if a particular data point was used in the training set of a model. Traditional studies on membership inference (Shokri et al., 2017; Yeom et al., 2018; Carlini et al., 2022; Wen et al., 2022) have predominantly focused on classifiers. An attacker can utilize losses or confidence scores as a metric. This is because data points from the training set typically exhibit lower losses or higher confidence scores than the unseen data points during inference due to overfitting. In a parallel development, recent works (Matsumoto et al., 2023; Duan et al., 2023; Wang et al., 2024; 2023) have extended membership inference to diffusion models. These methodologies involve introducing noise to a target image and subsequently verifying if the predicted noise aligns closely with the induced noise.

**Training Data Extraction.** Somepalli et al. (2022) demonstrate that diffusion models memorize a subset of their training data, often producing the training image verbatim. Building on this fact, Carlini et al. (2023) introduce a black-box data extraction attack designed for diffusion models. This approach involves generating a multitude of images and subsequently applying a membership inference attack to assess generation density. Notably, they observe that memorized prompts tend to produce nearly identical images across different seeds, leading to high density. This strategy bears resemblance to the pipeline used by Carlini et al. (2021), who successfully extract training data from large language models with over a billion parameters. Additionally, they discover that larger models are more susceptible to data extraction attacks compared to their smaller counterparts.

**Diffusion Memorization Mitigation.** Recent research by Daras et al. (2023) presents a method for training diffusion models using corrupted data. In their study, they demonstrate that their proposed training algorithm aids in preventing the model from overfitting to the training data. Their approach involves introducing additional corruption prior to the noising step and subsequently calculating the loss on the original input image. In a separate study, Somepalli et al. (2023b) delve into various mitigation strategies, with a focus on altering the text conditions. As a notable example, by inserting random tokens into the prompt or integrating random perturbations into the prompt embedding, they alleviate the memorization concern while preserving a high-quality generation output.

## 3 DETECT MEMORIZATION EFFICIENTLY

### 3.1 PRELIMINARY

We begin by defining the essential notation associated with diffusion models (Ho et al., 2020; Song & Ermon, 2020; Dhariwal & Nichol, 2021). For a data point $x_0$ drawn from the real data distribution $q(x_0)$, a forward diffusion process comprises a fixed Markov chain spanning $T$ steps, where each step introduces a predetermined amount of Gaussian noise. Specifically:

$$q(x_t|x_{t-1}) = \mathcal{N}(x_t; \sqrt{1 - \beta_t}x_t, \beta_t\mathbf{I}), \quad \text{for } t \in \{1, ..., T\}, \tag{1}$$

where $\beta_t \in (0, 1)$ is the scheduled variance at step $t$. The closed-form for this sampling is

$$x_t = \sqrt{\bar{\alpha}_t}x_0 + \sqrt{1 - \bar{\alpha}_t}\epsilon, \tag{2}$$

where, $\bar{\alpha}_t = \prod_{i=1}^{t}(1 - \beta_t)$.

In the reverse diffusion process, a Gaussian vector $x_T \sim \mathcal{N}(0, 1)$ is denoised to map to an image $x_0 \in q(x)$. At each denoising step, a trained noise-predictor $\epsilon_\theta$ anticipates the noise $\epsilon_\theta(x_t)$ that was added to $x_0$. Based on Eq. (2), the estimation of $x_0$ can be formulated as:

$$\hat{x}_0^t = \frac{x_t - \sqrt{1 - \bar{\alpha}_t}\epsilon_\theta(x_t)}{\sqrt{\bar{\alpha}_t}}. \tag{3}$$

Then, we can predict $x_{t-1}$ as:

$$x_{t-1} = \sqrt{\bar{\alpha}_{t-1}}\hat{x}_0^t + \sqrt{1 - \bar{\alpha}_{t-1}}\epsilon_\theta(x_t). \tag{4}$$

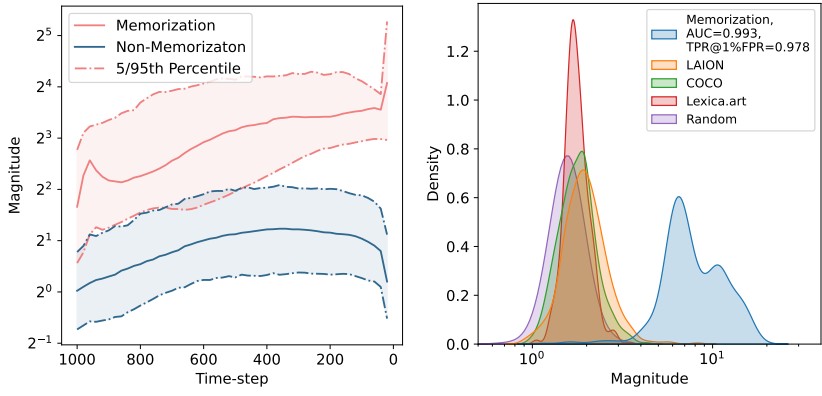

(a) Lineplot over generation steps      (b) Histogram over different datasets

Figure 2: Statistics of the magnitude of text-conditional noise predictions.

Text-conditional diffusion models, such as Stable Diffusion (Rombach et al., 2022), employ classifier-free diffusion guidance (Rombach et al., 2022) to steer the sampling process. Given a text prompt $p$, its embedding $e_p = f(p)$ is computed using a pre-trained CLIP text encoder $f(\cdot)$ (Radford et al., 2021; Cherti et al., 2023). In the reverse process, the conditional sampling adheres to Eq. (3) and Eq. (4), but the predicted noise $\epsilon_\theta(x_t)$ is changed to:

$$\epsilon_\theta(x_t, e_\emptyset) + s(\underbrace{\epsilon_\theta(x_t, e_p) - \epsilon_\theta(x_t, e_\emptyset)}_{\text{text-conditional noise prediction}}),$$

where, $e_\emptyset$ represents the prompt embedding of an empty string, and $s$ determines the guidance strength, controlling the alignment of the generation to the prompt. We refer to the term $\epsilon_\theta(x_t, e_p) - \epsilon_\theta(x_t, e_\emptyset)$ as the text-conditional noise prediction for future reference.

### 3.2 MOTIVATION

When provided with the same text prompt but different initializations, diffusion models can generate a diverse set of images. Conversely, when given different text prompts but the same initialization, the resulting images often display semantic similarities. These similarities include analogous layouts and color themes, as demonstrated in Appendix Fig. 6. Such a phenomenon might arise when the final generation remains closely tied to its initialization, and the textual guidance is not particularly dominant. This observation is consistent with findings from Wen et al. (2023b), suggesting that one can trace the origin to the initial seed even without knowing the text condition.

Interestingly, when it comes to memorized prompts, the initialization appears to be irrelevant. The generated images consistently converge to a specific memorized visualization. This behavior implies that the model might be overfitting to both the prompt and a certain denoising trajectory, which leads to the memorized image. Consequently, the final image deviates substantially from its initial state.

These insights provide a foundation for a straightforward detection strategy: scrutinizing the magnitude of text-conditional noise predictions. A smaller magnitude signals that the final image is closely aligned with its initialization, hinting that it is likely not a memorized image. On the other hand, a larger magnitude could indicate potential memorization. The correlation between magnitude and memorization is depicted in Fig. 2(a).

### 3.3 AN EFFECTIVE DETECTION METHOD

Following the intuition above, we introduce a straightforward yet effective detection method centered on the magnitude of text-conditional noise predictions. For a prompt embedding $p$ and a sampling step of $T$, the detection metric is defined as

$$d = \frac{1}{T} \sum_{t=1}^{T} \|\epsilon_\theta(x_t, e_p) - \epsilon_\theta(x_t, e_\emptyset)\|_2.$$

Table 1: Memorization detection results with AUC, TPR@1%FPR, and the running time of the method in seconds. In this table, "n" represents the number of generations per prompt.

| Method | 1st Step | First 10 Steps | Last Step |
|---|---|---|---|
| | AUC↑ / TPR@1%FPR↑ / Time in Seconds↓ | | |
| Density$_{\ell_2}$, n=4 | 0.520 / 0.012 / 0.810 | 0.652 / 0.225 / 5.314 | 0.659 / 0.288 / 9.904 |
| Density$_{\ell_2}$, n=16 | 0.506 / 0.000 / 3.570 | 0.656 / 0.175 / 24.78 | 0.676 / 0.271 / 40.66 |
| Density$_{\ell_2}$, n=32 | 0.510 / 0.000 / 8.092 | 0.664 / 0.175 / 59.43 | 0.681 / 0.266 / 81.44 |
| Density$_{SSCD}$, n=4 | 0.537 / 0.019 / 0.809 | 0.405 / 0.005 / 5.421 | 0.878 / 0.525 / 9.892 |
| Density$_{SSCD}$, n=16 | 0.515 / 0.000 / 3.186 | 0.375 / 0.000 / 21.02 | 0.934 / 0.523 / 39.55 |
| Density$_{SSCD}$, n=32 | 0.506 / 0.000 / 6.341 | 0.370 / 0.000 / 42.12 | 0.940 / 0.530 / 79.47 |
| **Ours**, n=1 | 0.960 / 0.760 / **0.199** | 0.989 / 0.944 / **1.866** | 0.989 / 0.934 / **9.584** |
| **Ours**, n=4 | 0.990 / 0.912 / 0.794 | 0.998 / 0.982 / 7.471 | 0.996 / 0.978 / 37.27 |
| **Ours**, n=32 | **0.996 / 0.954** / 1.606 | **0.999 / 0.988** / 14.96 | **0.998 / 0.986** / 74.75 |

Memorization is then identified if the detection metric falls beneath a tunable threshold $\gamma$.

In practice, we also find that even the detection metric of a single generation can provide a strong signal of memorization. Consequently, our method remains effective and reliable with the number of generations restricted to 1. In contrast, earlier studies, such as those examining generation density over a large number of generations (Carlini et al., 2023), require the simultaneous generation of multiple images, with some cases necessitating over a hundred generations. This might impose an extra computational burden on the service provider by generating more images than the user requested. Moreover, another method presented in (Somepalli et al., 2023a) identifies memorized prompts by directly comparing the generated images with the original training data. Unlike this method, our approach allows a third party to use the detection method without needing access to the large training dataset, thereby protecting training data privacy.

Another distinct advantage of our approach is its adaptability in calculating the detection metric. Strong detection does not mandate collecting the metric from all sampling steps. Based on our empirical findings, even when the metric is collated solely from the first step, reliable detection remains attainable. This efficiency enables model owners to identify memorized prompts promptly. By stopping generation early, they can then opt for post-processing, like declining the generated output or reinitializing the generation with corrective strategies in place.

### 3.4 EXPERIMENTS

**Experimental Setup.** To evaluate our detection method, we use 500 memorized prompts identified in Webster (2023) for Stable Diffusion v1 (Rombach et al., 2022), where the SSCD similarity score (Pizzi et al., 2022) between the memorized and the generated images exceeds 0.7. The memorized prompts gathered in Webster (2023) include three types of memorization: 1) matching verbatim: where the images generated from the memorized prompt are an exact pixel-by-pixel match with the original paired training image; 2) retrieval verbatim: the generated images perfectly align with some training images, albeit paired with different prompts; 3) template verbatim: generated images bear a partial resemblance to the training image, though variations in colors or styles might be observed.

Additionally, we use another 2,000 prompts, evenly distributed from sources LAION (Schuhmann et al., 2022), COCO (Lin et al., 2014), Lexica.art (Santana, 2022), and randomly generated strings. For this set of prompts, we assume they are not memorized by the model. All generations employ DDIM (Song et al., 2020) with 50 inference steps.

In our comparison, we use the detection method from Carlini et al. (2023) as a baseline. This method determines memorization by analyzing generation density, computed using the pairwise $\ell_2$ distance between non-overlapping tiles. While Carlini et al. (2023) utilizes the $\ell_2$ distance in pixel space, we introduce an additional baseline that calculates the distance in the SSCD feature space (Pizzi et al., 2022). This adjustment is inspired by Somepalli et al. (2022), who underscore the effectiveness of

SSCD. As a deep learning-informed distance metric, SSCD offers enhanced resilience to particular augmentations, like color shift — a critical advantage when the training image is only partially memorized.

We use the area under the curve (AUC) of the receiver operating characteristic (ROC) curve and the True Positive Rate at the False Positive Rate of $1\%$ (TPR@1%FPR) as metrics. Meanwhile, we report the running time in seconds with a batch size of $4$ on a single NVIDIA RTX A6000.

**Results.** In Fig. 2(b), we display a density plot comparing the detection metrics for the memorized prompts against the non-memorized ones, calculated over $50$ steps with $4$ generations per prompt. The distribution of memorized prompts is bimodal. This dichotomy stems from the fact that the template verbatim scenario often exhibits a slightly smaller metric than the matching verbatim scenario, given that the memorization occurs only partially.

In Table 1, we highlight the balance between the precision and efficiency of our proposed method. Our method is able to achieve very strong detection performance. When generating $32$ images and using the metrics from the first $10$ steps, our method is able to achieve an AUC of $0.999$ and TPR@1%FPR of $0.988$. Remarkably, even when operating with a single generation, our method can achieve TPR@1%FPR of $0.760$ from the very first step within merely $0.2$ seconds. This feature provides a significant advantage in terms of time and computational resource savings, allowing model operators the flexibility to terminate generation early if necessary. In contrast, the baseline methods show noticeably reduced detection capability. In particular, the baseline methods can only achieve relatively high detection accuracy when generating more than $16$ images and relying on the image generation from the final step, where it requires at least $40$ seconds. Yet, in real-world applications, service providers like *Midjourney* or *DALL-E 2* (Ramesh et al., 2022) typically generate a mere $4$ images concurrently for each prompt.

Interestingly, our method surpasses the baselines in speed even when using the metric with equivalent generations and steps. This superiority emerges since our method doesn't need to decode latent noise into image space and perform subsequent calculations.

## 4 MITIGATE MEMORIZATION

### 4.1 A STRAIGHTFORWARD METHOD TO DETECT TRIGGER TOKENS

As observed by Somepalli et al. (2022), certain words or tokens in memorized prompts play a significant influence on the generation process. Even when only these specific "trigger tokens" are present in the prompt, the memorization effect remains evident. One potential approach to identify these trigger tokens involves probing with various n-gram combinations to discern which combinations induce memorization. However, this heuristic becomes notably inefficient, particularly when the prompt contains a vast number of tokens. Our earlier observations offer a more streamlined method for discerning the significance of each token in relation to memorization: by checking the magnitude of the change applied to each token while minimizing the magnitude of text-conditional noise prediction. A token undergoing substantial change suggests its crucial role in steering the prediction; conversely, a token with minimal change is less important.

Given a prompt embedding $e$ of prompt $p$ with $N$ tokens, we form the objective of the minimization problem as:

$$\mathcal{L}(x_t, e) = \|\epsilon_\theta(x_t, e) - \epsilon_\theta(x_t, e_\emptyset)\|_2. \tag{5}$$

We then determine the significance score for each token at position $i \in [0, N-1]$ as:

$$\text{SS}_{e^i} = \frac{1}{T} \sum_{t=1}^{T} \|\nabla_{e^i} \mathcal{L}(x_t, e)\|_2.$$

In Fig. 3, we display generations with top-2 significant tokens highlighted. The green arrow in the figure emphasizes that altering these significant tokens can substantially diminish the memorization effect. Some trigger tokens, including symbols or seemingly trivial words, are challenging to identify manually. Consequently, this insight offers model owners a practical tool: advising users to rephrase or exclude the trigger tokens before initiating another generation. In contrast, as indicated

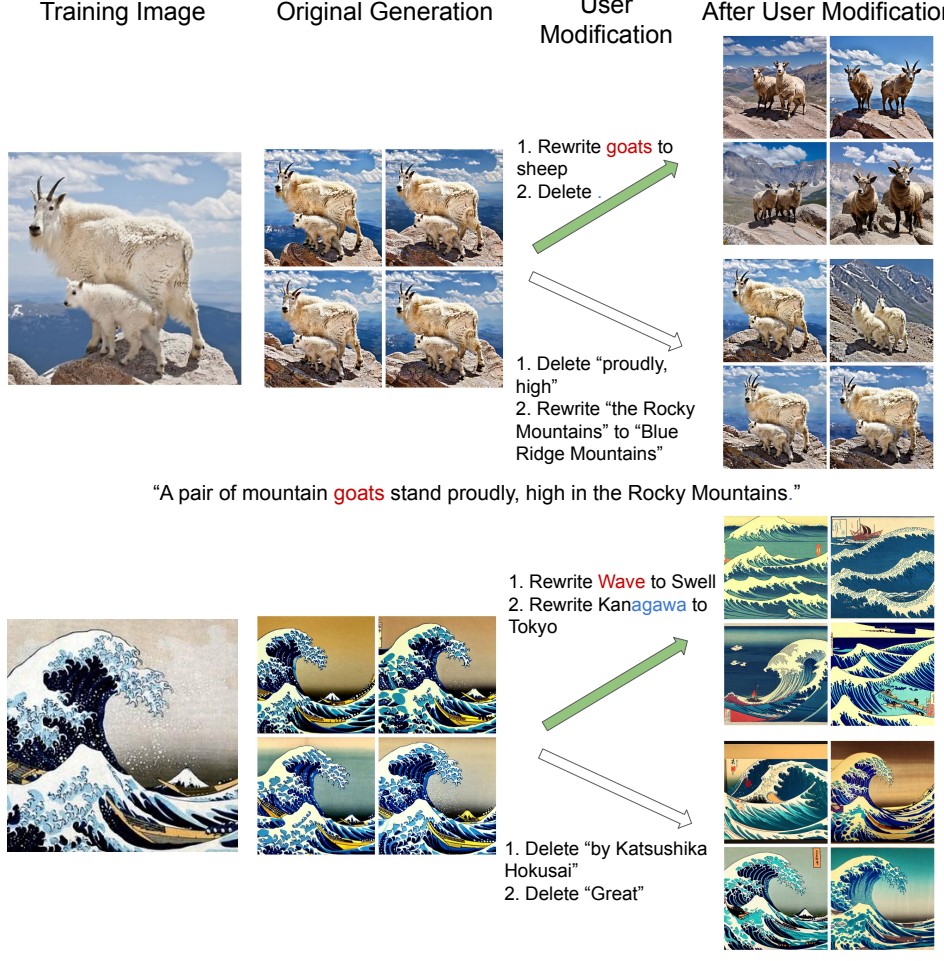

"A pair of mountain goats stand proudly, high in the Rocky Mountains."

"A painting of the Great Wave off Kanagawa by Katsushika Hokusai"

Figure 3: By modifying the trigger tokens, memorization can be effectively mitigated. The significance score for each token is illustrated in a histogram in Appendix Fig. 7. The two most significant tokens are highlighted in **red** and **blue**. A green arrow indicates modifications made to the top-2 tokens, while a white arrow represents changes to less significant tokens.

by the white arrow, alterations to the less significant tokens fail to effectively counter the memorization effect, even when extensive changes are made. Even by renaming the play or removing the artist's name, memorization remains evident.

## 4.2 AN EFFECTIVE INFERENCE-TIME MITIGATION METHOD

A direct approach to mitigation without any supervision is to adjust the prompt embedding by minimizing Eq. (5). Optimizing over all time steps is computationally intensive. However, we observe that minimizing the loss at the initial time step indirectly results in smaller magnitudes in subsequent time steps, effectively mitigating memorization. Thus, a perturbed prompt embedding, $e^*$, is obtained at $t = 0$ by minimizing Eq. (5). We also apply early stopping once the loss reaches a target value, $l_{\text{target}}$, to keep the embedding close to the original meaning.

## 4.3 AN EFFECTIVE TRAINING-TIME MITIGATION METHOD

During the training of diffusion models, memorization often arises with duplicate data points due to overfitting (Carlini et al., 2023; Somepalli et al., 2023b). Building on our earlier observation, if the model overfits or closely memorizes a data point, the magnitude of the text-conditional noise

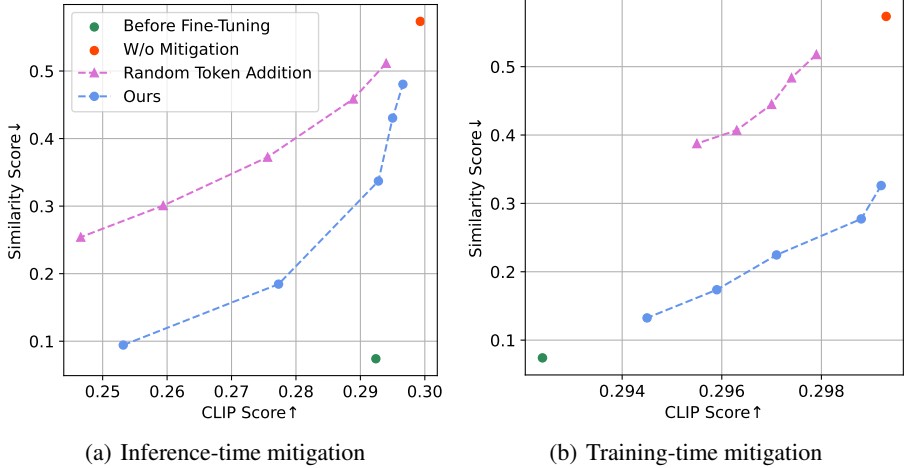

(a) Inference-time mitigation          (b) Training-time mitigation

Figure 4: Mitigation results. A lower similarity score suggests reduced memorization, whereas a higher clip score denotes a better alignment between the generation and the prompt.

prediction might exceed typical values. Therefore, a straightforward mitigation method is to exclude the sample from the current mini-batch if this magnitude surpasses a predetermined threshold, $\tau$, thereby not computing the loss on that sample. Given that the model has previously seen the sample during training, this exclusion is unlikely to significantly impact model performance.

However, this method introduces an additional computational cost during training. To compute the text-conditional noise prediction, an extra forward pass for $\epsilon_\theta(x_t, e_\emptyset)$ is needed. During typical training, only $\epsilon_\theta(x_t, e_p)$ is computed. Empirically, this results in an approximately $10\%$ increase in training time.

## 4.4 EXPERIMENTS

**Experimental Setup.** To evaluate the effectiveness of mitigation strategies, we adopt a setup similar to that in Somepalli et al. (2023b). Specifically, we fine-tune the Stable Diffusion model using 200 LAION data points, each duplicated 200 times, to serve as memorized prompts. In addition, we introduce $120,000$ distinct LAION data points to ensure that the model retains its capacity for generalization.

For performance metrics, we compute the SSCD similarity score (Pizzi et al., 2022; Somepalli et al., 2023b) to gauge the degree of memorization by comparing the generation to the original image. Additionally, the CLIP score (Radford et al., 2021) is used to quantify the alignment between the generation and its corresponding prompt. Our experiments encompass 5 distinct fine-tuned models, each embedded with different memorized prompts, and the results are averages over 5 runs with different random seeds.

In our evaluation of the proposed method, we test 5 distinct target losses $l_{\text{target}}$, ranging from 1 to 5, for inference-time mitigation. We use Adam optimizer (Kingma & Ba, 2014) with a learning rate of $0.05$ and at most 10 steps. Simultaneously, we investigate 5 different thresholds $\tau$, spanning from 2 to 6, for training-time mitigation. For comparison, we use the most effective method from (Somepalli et al., 2023b), random token addition (RTA), as the baseline, which inserts $1, 2, 4, 6,$ or 8 random tokens to the prompt.

**Results.** In Fig. 4(a), we present the results of our inference-time mitigation, while Fig. 4(b) details the outcomes for training-time mitigation. Our proposed techniques successfully mitigate the memorization effect and, importantly, offer a more favorable CLIP score trade-off compared to RTA. Higher target losses $l_{\text{target}}$ or thresholds $\tau$ tend to enhance the model's alignment with the prompt but can result in a less pronounced mitigation effect. In practice, model owners can select the optimal hyperparameter based on their desired balance between mitigation efficacy and generation alignment.

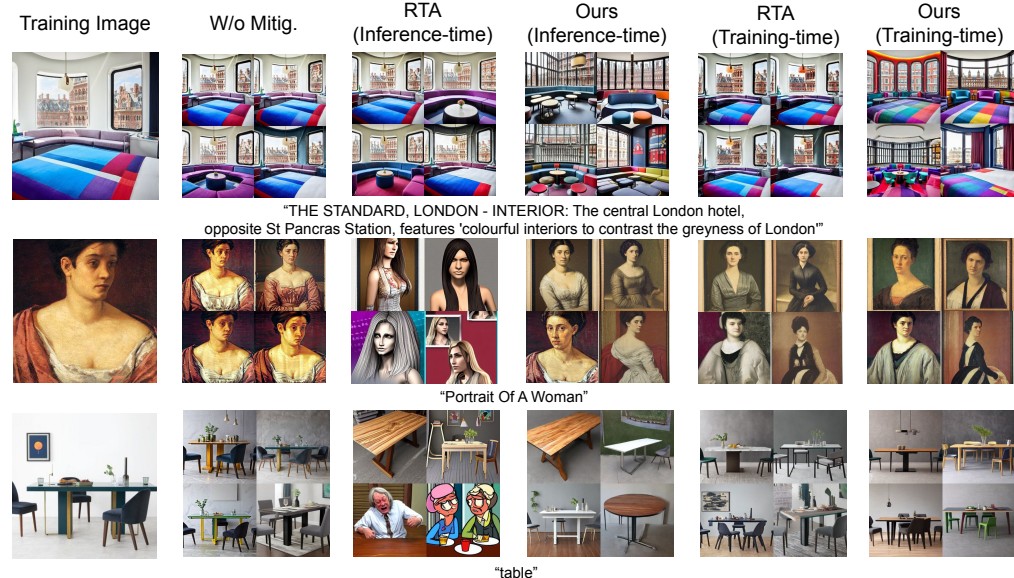

Figure 5: Mitigation results with different mitigation strategies during inference and training phase.

In Fig. 5, we display a selection of qualitative results, setting $l_{\text{target}} = 3$ and $\tau = 4$ for our method, while adding 4 random tokens as a baseline strategy. Our proposed strategy is effective in mitigating the memorization effect while also ensuring that the generated content aligns closely with the prompt. In contrast, the baseline method often struggles to counteract memorization and occasionally produces images with undesirable additions.

## 5 LIMITATIONS AND FUTURE WORK

Our detection strategy employs a tunable threshold to detect memorized prompts. This requires model owners to first compute metrics over non-memorized prompts and then select an empirical threshold based on a predetermined false positive rate. However, the outcomes generated by this detection approach lack interpretability. In the future, the development of a method producing interpretable p-values could significantly assist model owners by providing a confidence score that quantifies the likelihood of memorization, thereby augmenting the transparency and trustworthiness of the detection process.

Our proposed mitigation strategies effectively tackle the memorization problem, albeit with minor computational overheads for model owners. The inference-time mitigation approach requires additional GPU memory due to the optimization process, and in practice, it takes at most 6 seconds with our setup. On the other hand, the training-time mitigation extends the training period by roughly 10%. For context, while naive fine-tuning in our tests takes approximately 9 hours, our mitigation strategy extends this to about 10 hours. However, we argue that these added costs are reasonable given the significance of protecting training data privacy and intellectual property. Furthermore, when combined with our detection method, model owners need only deploy the inference-time mitigation when a memorized prompt is detected, thereby minimizing its use.

## 6 CONCLUSION

In this paper, we introduced a new approach to detect memorization in diffusion models by leveraging the magnitude of text-conditional noise predictions. Remarkably, our approach attains high precision even when using a limited number of generations per prompt and a limited number of sampling steps. Furthermore, we provide an explanatory tool to indicate the significance score of individual tokens in relation to memorization. To conclude, our paper presents both inference-time and training-time mitigation strategies. These not only effectively address the memorization concern but also maintain the superior generative performance of the model.

## 7 REPRODUCIBILITY STATEMENT

We have detailed all essential hyperparameters used in our experiment setup in the main body. Our experiments were conducted using widely available computing resources and open-source software. All referenced models and datasets in this paper are publicly accessible. Furthermore, we have included the code necessary to reproduce our results in the supplementary material.

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

## A APPENDIX

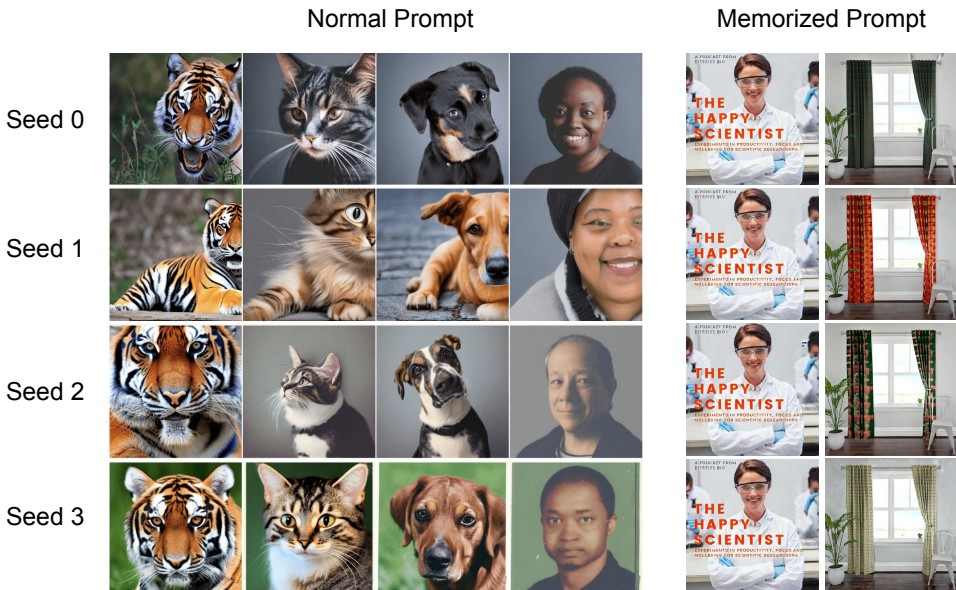

Figure 6: Different seeds vs. different prompts.

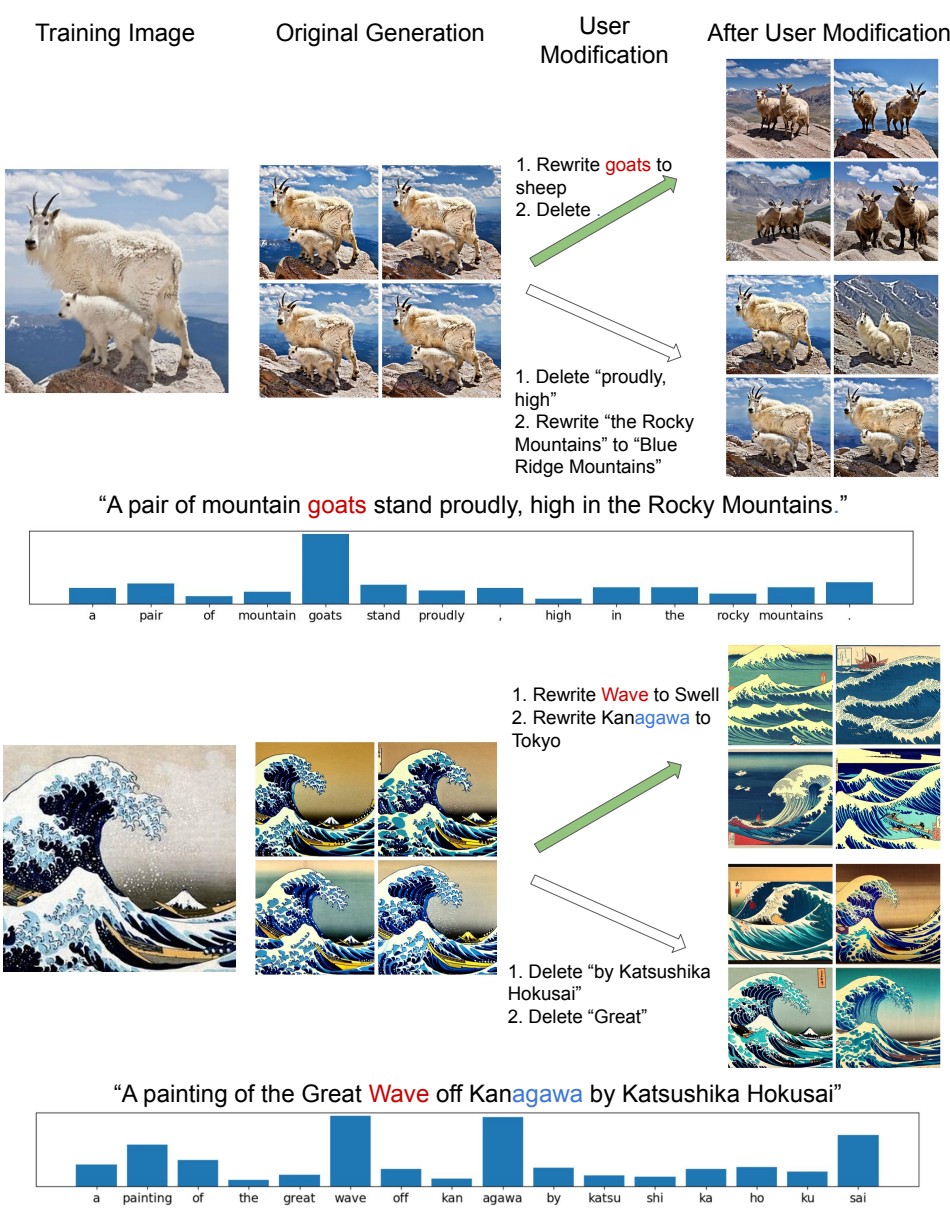

Figure 7: By modifying the trigger tokens, memorization can be effectively mitigated. The significance score for each token is illustrated in a histogram in Appendix Fig. 7. The two most significant tokens are highlighted in **red** and **blue**. A green arrow indicates modifications made to the top-2 tokens, while a white arrow represents changes to less significant tokens.

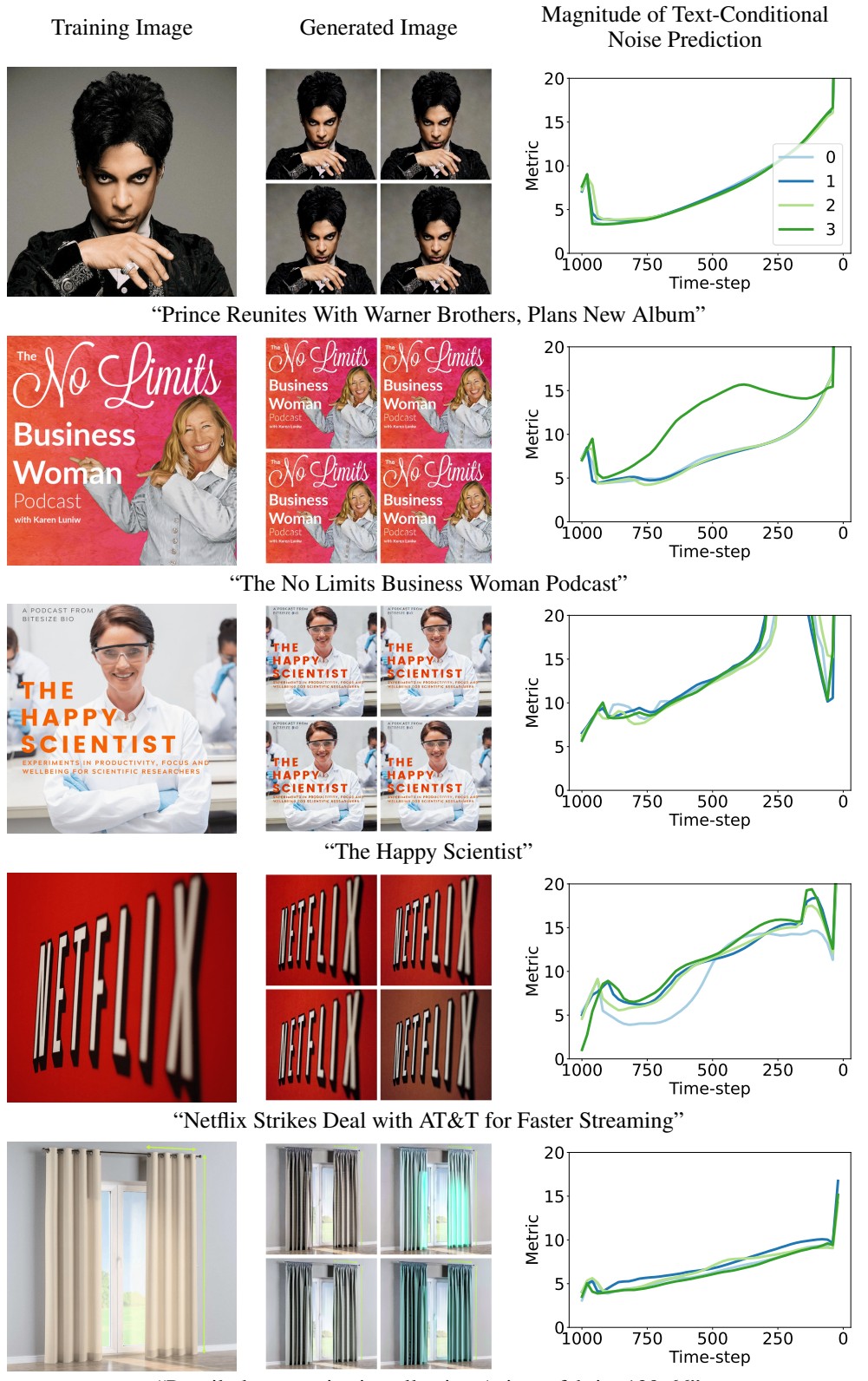

Figure 8: Memorization generation. We display the magnitude of text-conditional noise prediction at each time-step, as described in Section 3.3, for all four generations distinctly (with 4 different random seeds) for each prompt. The metric typically indicates a higher value when memorization occurs.

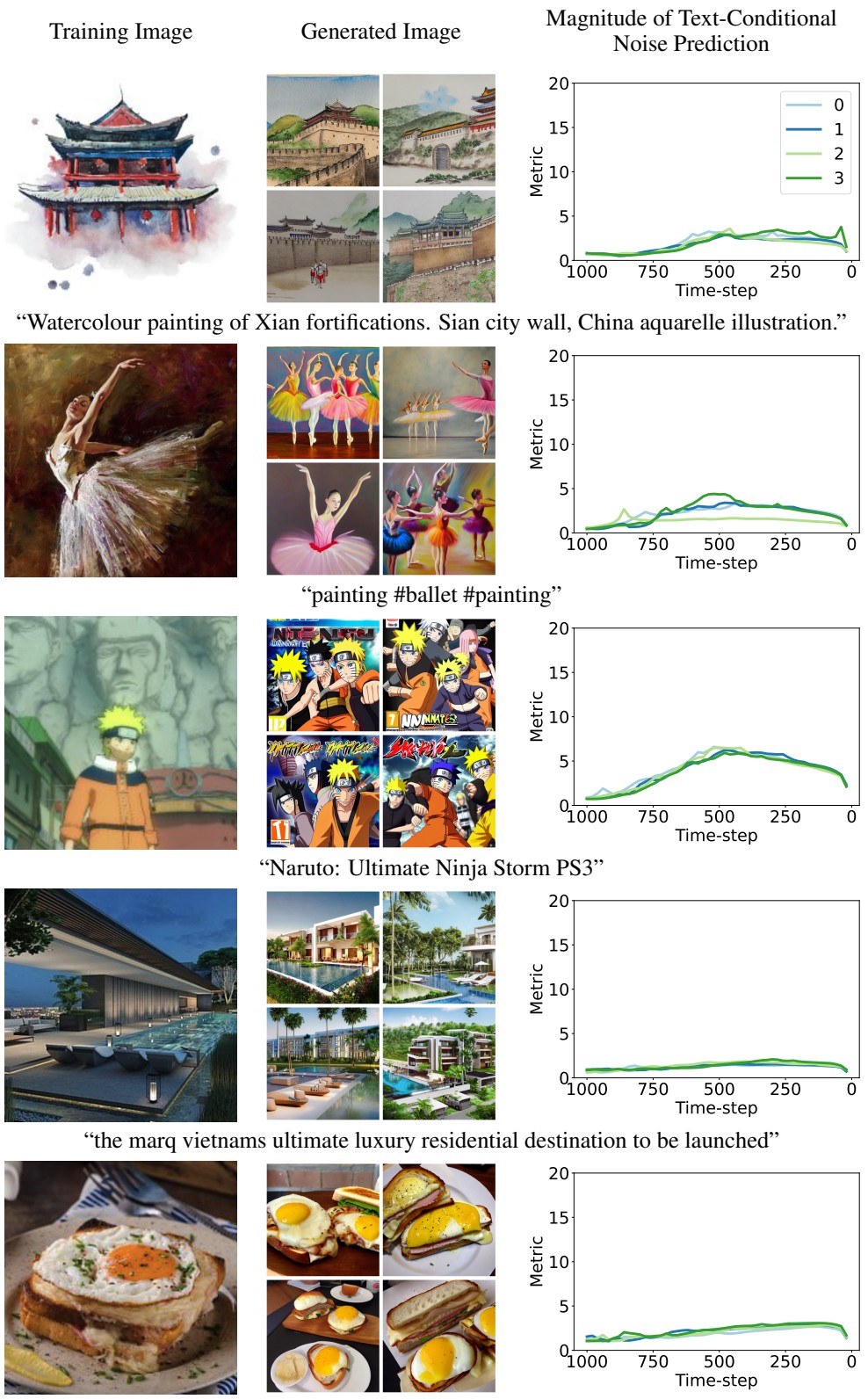

Figure 9: Non-memorization generation. We display the magnitude of text-conditional noise prediction at each time-step, as described in Section 3.3, for all four generations distinctly (with 4 different random seeds) for each prompt. The non-memorized generations consistently exhibit significantly lower metric values.

