# OpenReview forum: "Detecting, Explaining, and Mitigating Memorization in Diffusion Models"
_ICLR.cc/2024/Conference — ICLR 2024 oral_

### Official Review · Reviewer_hCP2 · 2023-10-30

**Soundness:** 2 fair
**Presentation:** 3 good
**Contribution:** 3 good
**Rating:** 8
**Confidence:** 2

**Summary:**

The paper discusses memorization issue of diffusion models used in generating images from text prompts. Text-conditional diffusion models employ a pre-trained text encoder, which controls the alignment of image generation to a given prompt. Observations reveal that diffusion models generate diverse images for the same text prompt but different initializations. However, with different prompts but identical initializations, the images display similarities. When the model uses memorized prompts, the output tends to be consistent, hinting at overfitting. One of the significant findings is the relationship between the magnitude of text-conditional noise predictions and the chances of an image being memorized. This paper proposed a method, allowing early detection of memorized prompts, potentially saving computational resources. The effectiveness of this method is further discussed in an experimental setup, where it surpasses other baseline methods in speed and accuracy. Additionally, the concept of \emph{trigger tokens} is introduced. These are specific words or tokens in a prompt that have a significant impact on the generation process. A technique is provided to identify these tokens, allowing users to modify them to mitigate memorization. In summary, the paper explores the intricacies of diffusion models in generating images from text prompts and provides methods to detect and counter memorized image generation.

**Strengths:**

1. The paper introduces a novel method to efficiently detect memorization in diffusion models by scrutinizing the magnitude of text-conditional noise predictions. This method is both computationally efficient and does not require multiple generations or access to the original training data, ensuring data privacy and reducing computational overhead.

2. This work offers an automated approach to identify specific "trigger tokens" in memorized prompts that have a significant influence on the generation process. Instead of manual identification or experimentation with various token combinations, which can be cumbersome and inefficient, the paper's method assesses the change applied to each token while minimizing the magnitude of text-conditional noise prediction. This innovative approach provides model owners with a practical tool to advise users on how to modify or omit these trigger tokens, which can significantly mitigate the effects of memorization.

3. The authors introduce mitigation methods that cater to both inference and training phases. For inference, a perturbed prompt embedding is suggested, achieved by minimizing the magnitude of text-conditional predictions. During training, potentially memorized image-text pairs can be screened out based on the magnitude of text-conditional predictions. These methods not only address the concerns of memorization but also ensure a more consistent alignment between prompts and generations. The experiments conducted, as per the paper's context, seem to support the efficacy of these strategies when benchmarked against baseline mitigation methods.

**Weaknesses:**

1. While the mitigation strategies aim to reduce memorization, it's unclear what impact they might have on the overall performance of the model. Often, there's a trade-off between reducing a particular behavior and maintaining high performance. If these mitigation strategies significantly impair the model's utility, it might deter their adoption.

2. As stated in the paper, a weakness of the proposed method is the lack of interpretability in the detection strategy of memorized prompts. The current approach requires the model owners to select an empirical threshold based on a predetermined false positive rate, but the outcomes generated lack clear interpretability. This lack of clarity can make it difficult for model owners to fully understand and trust the detection process. The authors acknowledge that developing a method that produces interpretable p-values could significantly assist model owners by providing a confidence score quantifying the likelihood of memorization.

3. Advising users on modifying or omitting trigger tokens might be effective in theory, but in practice, it could be cumbersome. Users might need to understand what these tokens are, why they need to modify them, and how they affect the output. This could make the user experience less intuitive, especially for those unfamiliar with the inner workings of AI models.

4. The paper assumes that all prompts can be modified or that users will be willing to modify them. In real-world scenarios, some prompts might be non-negotiable, and changing them might not be an option.

5. While the paper suggests that the method is computationally efficient, implementing the strategies during the training and inference phases might still introduce computational or operational overheads for model owners.

**Questions:**

1. Is there any way to quatify memorization in the diffusion models that the future method could use to benchmark? It might be good to have a discussion in this direction.

2. Are certain tokens more susceptible to triggering memorization than others? How were these trigger tokens identified, and is there a taxonomy or classification for them?

3. Were there any adversarial tests done to ascertain if an attacker could still exploit the memorization tendencies, even after applying the proposed mitigation strategies?

---

> ### Author Response · Authors · 2023-11-21
> **Response to Reviewer hCP2 (part 1/2)**
>
> We sincerely appreciate your valuable feedback and the time you've dedicated to providing it. Below, we address specific points you raised:
>
> > While the mitigation strategies aim to reduce memorization, it's unclear what impact they might have on the overall performance of the model. Often, there's a trade-off between reducing a particular behavior and maintaining high performance. If these mitigation strategies significantly impair the model's utility, it might deter their adoption.
>
> Yes, as shown in Figure 4, there is a small degradation in CLIP score for inference-time mitigation, but not for training-time mitigation. However, we want to emphasize that in practice, the model owner can deploy detection and mitigation strategies at the same time. Therefore, the mitigation will be only applied to memorized prompts, and, in general case, the model performance will not be impaired.
>
> Meanwhile, compared to the copyright issue or privacy issue of outputting memorized training data, we believe it is still beneficial to the company with a little drop in CLIP score in the memorization case.
>
> > User interaction
>
> We acknowledge your concerns regarding the potential complexity of modifying prompts for some users. However, prompt interaction might be a potential feature. A notable example of this is OpenAI's DALL-E 3, which allows users to iteratively refine their prompts to achieve the desired results. This advanced level of user interaction illustrates both the practicality and the growing trend of user familiarity with such interactive processes. We anticipate that, in the future, users will become increasingly adept at interacting with the models.
>
> Addressing memorization is crucial for users. If the model's outputs contain copyrighted images, users who further utilize these images might also encounter legal issues. Additionally, for memorized prompts, the generated content is often limited to the memorized images. Therefore, modifying the prompt is beneficial for the users to achieve more diverse generations.
>
> Furthermore, for situations where users find it cumbersome to modify prompts or encounter hard-to-alter tokens, the automatic inference-time mitigation method proposed by our paper could be an alternative. This option provides flexibility, catering to different user preferences and capabilities.
>
> Overall, our proposed method offers a new perspective on designing user interfaces and addressing the issue of memorization. We believe that our approach contributes valuable insights for future developments in the field.
>
> > While the paper suggests that the method is computationally efficient, implementing the strategies during the training and inference phases might still introduce computational or operational overheads for model owners.
>
> We agree that there is a small computational overhead for model owners. Specifically, as mentioned in our paper, our detection method does not require additional computation since the text-conditional noise prediction is calculated during inference. Meanwhile, the inference-time mitigation incurs around 6 seconds for optimization, and a 10% increase in time for training-time mitigation. These costs are negligible compared to potential lawsuits due to memorization. Also, we believe that there is no free lunch to mitigate memorization. Therefore, a minor cost to achieve effective memorization reduction is acceptable.
>
> > Is there any way to quatify memorization in the diffusion models that the future method could use to benchmark? It might be good to have a discussion in this direction.
>
> Yes, in our work, we adopt the definition of memorization from previous research [1], where we measure the SSCD similarity between training and generated images using the memorized prompts dataset from [2].
>
> However, we acknowledge the potential for developing a more comprehensive benchmark in the future. For instance, categorizing memorization into different types of legal copyright infringement could offer more practical insights. This approach would allow for a comparative analysis of methods under various memorization categories. Although establishing such a benchmark may be labor-intensive, we believe it is crucial and advantageous for advancing the development of reliable generative AI.

---

> > ### Author Response · Authors · 2023-11-21
> > **Response to Reviewer hCP2 (part 2/2)**
> >
> > > Are certain tokens more susceptible to triggering memorization than others? How were these trigger tokens identified, and is there a taxonomy or classification for them?
> >
> > This problem was well studied by [1]. The susceptiblility of triggering memorization really depends on the repetition of a certain word associated with the same image in the training data. However, if under the same repetition rate, [1] found in generating more specific captions are easier to memorize due to their uniqueness.
> >
> > > Were there any adversarial tests done to ascertain if an attacker could still exploit the memorization tendencies, even after applying the proposed mitigation strategies?
> >
> > We have yet to discover any adversarial methods capable of evading our proposed detection and mitigation strategy. One trial we tried was to utilize PEZ[3] to optimize a new prompt with the CLIP model. However, we observed that significant text-conditional noise prediction occurs whenever memorization is involved, affirming the effectiveness of our detection and mitigation approaches.
> >
> > We hope our response can solve your concern regarding our paper. Please let us know if you have any more questions.
> >
> > [1] Somepalli, G., Singla, V., Goldblum, M., Geiping, J., & Goldstein, T. (2023). Understanding and Mitigating Copying in Diffusion Models. arXiv preprint arXiv:2305.20086.
> >
> > [2] Webster, R. (2023). A Reproducible Extraction of Training Images from Diffusion Models. arXiv preprint arXiv:2305.08694.
> >
> > [3] Wen, Y., Jain, N., Kirchenbauer, J., Goldblum, M., Geiping, J., & Goldstein, T. (2023, November). Hard prompts made easy: Gradient-based discrete optimization for prompt tuning and discovery. In Thirty-seventh Conference on Neural Information Processing Systems.

---

> ### Comment · Reviewer_hCP2 · 2023-11-22
> **Thanks for the detailed rebuttal**
>
> I thank the authors for preparing the detailed rebuttal and addressing my concerns. I have already raised my rating to accept.

---

### Official Review · Reviewer_VQfz · 2023-10-31

**Soundness:** 3 good
**Presentation:** 3 good
**Contribution:** 3 good
**Rating:** 8
**Confidence:** 3

**Summary:**

## Summary


Paper studies the problem of memorization in diffusion models. These models sometimes simply reproduce images from their training set which could present legal challenges for model owners
	- One real life examples of this in Midjourney, which had to ban prompts with the substring "Afghan" to avoid generating images reminiscent of the renowned copyrighted photograph of the "Afghan girl"
	-
- In this work, the authors introduce a metric to detect such memorized prompts based on the magnitude of "text-conditional predictions". Memorized prompts tend to have a higher magnitude than non-memorized prompts.
	- Extending this the authors also devise a strategy to highlight the influence of each token in the prompt in driving memorization. This is done by evaluating the change in gradient for every token when minimizing the "text-conditional prediction" magnitude. This gives the relative important of each token to memorization.
	- This can be used to provide feedback to prompt designers to omit, modify these pivotal trigger tokens in their prompt.
	- Stable diffusion uses classifier-free guidance to steer the sampling diffusion process. During the reverse diffusion process the noise part of the original equation is modified to minimize distance from  the embedding of the text computed using a pre-trained CLIP encoder. This difference term is referred to as the "text-conditional noise prediction". The metric proposed in the paper is defined as the L2 norm of "text-conditional noise" term divided by the number of sampling steps
		- A smaller magnitude for this term signifies the final image is closely aligned its initialization
Baseline & Dataset:
- The authors use the 500 memorized prompts from Webster 2023 for stable diffusion v1 where SSCD similarity score between memorized and generated images exceeds 0.7
- The detection method from Carlini 2023 is used as baseline
- They also use an additional baseline where instead of using L2 distance like Carlini 2023 they replace it with distance in the SSCD feature space.

**Strengths:**

### Strengths/Weaknesses

- The two advantages of using the proposed metric are
	- It doesn't need access to training data which some of the previous methods do
	- Even if the metric is collated solely from first step, reliable detection is possible.
- Results indicate that the method obtains a high detection score with an AUC of 0.999 with small latency.

**Weaknesses:**

See above

**Questions:**

## Questions/Clarifications

- How is the metric computed with multiple generations in Table 1?
- In Table 1, what does the column First 10 steps indicate, is it the AUC, TPR values calculated with the average of metric values for the first 10 steps of the diffusion process?
- What is meant by the following sentence in the "An effective inference-time mitigation method"
	- "Thus, a perturbed prompt embedding e* is obtained as t=0 by minimizing Eq (5)" - Is this minimization done via gradient descent, what is the data on which this minimization is performed?
	- Was this done on the 200 LAION data points? If yes, what is Figure 4(a) and 4(b) plotted over?

---

> ### Author Response · Authors · 2023-11-21
> **Response to Reviewer VQfz**
>
> We sincerely appreciate your valuable feedback and the time you've dedicated to providing it. Below, we address specific points you raised:
>
> > How is the metric computed with multiple generations in Table 1?
>
> For each prompt, the final metric is calculated as an average across all generations.
>
> > In Table 1, what does the column First 10 steps indicate, is it the AUC, TPR values calculated with the average of metric values for the first 10 steps of the diffusion process?
>
> Your interpretation is correct. It is calculated based on the average metric obtained during the initial 10 steps of the denoising process.
>
> > Clarification of "An effective inference-time mitigation method"
>
> We initiate the process with the original prompt embedding and employ gradient descent using the Adam optimizer. The objective function for this optimization is detailed in Equation 5 of our paper. It is crucial to note that this optimization process does not involve any data and is conducted independently for different prompts.
>
> Regarding Figures 4(a) and 4(b), the results were derived from an evaluation of 5 x 200 data points from the LAION dataset, with '5' representing five different random seeds. The detailed experiment setting can be found in section 4.4.
>
> We appreciate your insights pointing out ambiguities in our paper. We will address these issues to enhance clarity in the future version. Please let us know if you have any more questions.

---

### Official Review · Reviewer_Mwfj · 2023-11-01

**Soundness:** 4 excellent
**Presentation:** 4 excellent
**Contribution:** 4 excellent
**Rating:** 8
**Confidence:** 4

**Summary:**

The paper addresses recent breakthroughs in diffusion models, particularly focusing on their image generation capabilities. It highlights a significant issue where some outputs from these models are mere replications of training data, posing legal challenges, especially when the content includes proprietary information. The authors propose a method for detecting memorized prompts by examining the magnitude of text-conditional predictions. This method integrates seamlessly into existing sampling algorithms and provides high accuracy from the first generation step with a single generation per prompt.

**Strengths:**

1. The paper introduces a straightforward yet effective technique for detecting memorized prompts, which is a significant contribution to enhancing the reliability of diffusion models.
2. Mitigation Strategies: The paper proposes two practical strategies for mitigating memorization - minimization during inference and filtering during training. These strategies effectively balance counteracting memorization while maintaining high generation quality.

**Weaknesses:**

1. Clarifying the Concept of Memorization: Could you provide a clear definition of what constitutes memorization in this context? Does it require an exact match between the generated and training images? For instance, if there's a slight variation, such as a difference of 10 pixels from the original image in the training dataset, would that still be considered memorization?

**Questions:**

1. Exploring Applications for Memorization Detection: I'm interested in understanding the practical uses of detecting memorized images. What are some key scenarios or fields where identifying such images is particularly important or beneficial?

---

> ### Author Response · Authors · 2023-11-21
> **Response to Reviewer Mwfj**
>
> We sincerely appreciate your valuable feedback and the time you've dedicated to providing it. Below, we address specific points you raised:
>
> > Clarifying the Concept of Memorization: Could you provide a clear definition of what constitutes memorization in this context? Does it require an exact match between the generated and training images? For instance, if there's a slight variation, such as a difference of 10 pixels from the original image in the training dataset, would that still be considered memorization?
>
> Following [1], we employ the SSCD similarity score to measure memorization. As indicated in our experimental setup, for all memorized prompts from [2], the SSCD similarity score between the memorized and generated images is above 0.7.
>
> [1] and we advocate for the use of SSCD similarity over pixel space distance as a more reliable measure. This is especially relevant in cases where the generated image partially replicates the training image. SSCD focuses on structural and content similarities, not just pixel-level accuracy. This approach offers a more nuanced and context-sensitive evaluation of memorization.
>
> > Exploring Applications for Memorization Detection: I'm interested in understanding the practical uses of detecting memorized images. What are some key scenarios or fields where identifying such images is particularly important or beneficial?
>
> When using training datasets that are not as meticulously curated as, for example, the LAION dataset, there is a significant risk of including copyrighted or private images. This underscores the importance of detecting and preventing memorized images, particularly in two primary scenarios:
> 1. copyrighted images: When a model generates an image that is an exact replica of a copyrighted image, the owner of that copyright may have grounds to sue the model owner. This situation can result in financial losses for the model owner.
> 2. private images: The situation becomes more critical if the memorized image contains private data, like a patient's CT scan. Unauthorized reproduction of such private images can result in severe breaches of privacy, leading to legal and ethical issues.
>
> Given these scenarios, it is evident that the need for an effective memorization detection tool goes beyond these examples. There are likely other scenarios where such a tool would be crucial in managing the potential risks.
>
> We hope our response can solve your concern regarding our paper. Please let us know if you have any more questions.
>
> [1] Somepalli, G., Singla, V., Goldblum, M., Geiping, J., & Goldstein, T. (2022). Diffusion Art or Digital Forgery? Investigating Data Replication in Diffusion Models. 2023 IEEE/CVF Conference on Computer Vision and Pattern Recognition (CVPR), 6048-6058.
>
> [2] Webster, R. (2023). A Reproducible Extraction of Training Images from Diffusion Models. arXiv preprint arXiv:2305.08694.

---

### Official Review · Reviewer_5xmi · 2023-11-10

**Soundness:** 4 excellent
**Presentation:** 3 good
**Contribution:** 4 excellent
**Rating:** 8
**Confidence:** 4

**Summary:**

This paper builds up on previous work by Somepalli et al. on detecting and mitigating memorization in diffusion models. The authors use a very neat observation that in the case of text-guided diffusion models, the actual text prompt is very important for the final generated image. In particular, as seen by Wen et al., in the cases when the text prompt is important, irrespective of the initialization, the model would typically converge to the same image. However, in other cases, the initialization can change the model generation a lot. Using this insight, they detect memorization by evaluating the impact of the text on the model generation. The work is then further solidified by using this both as a detection and mitigation measure, surpassing past works not only by efficiency but also by performance.

**Strengths:**

1. This work builds on a very simple and clever observation that the impact of text prompt on the generation by a diffusion model can be used for detecting if a particular generated image was memorized.
2. The method is extremely fast and can even detect memorization with a single step. Further, it is much better than past works, both l2, and SSCD metrics in terms of the AUC and the true positive rate at 1% false positive rate.
3. The proposed mitigation strategies at inference time are very interesting and much more performant than the previously discussed baseline of random token addition. In particular, the method and the insight naturally offer a way of understanding which tokens were responsible for memorization and can be removed appropriately.
4. In the case of training time mitigation, the authors see significant improvement in the model performance as opposed to when you're doing random token addition.
5. Overall, this paper is a very enjoyable read and a strong work in the field of memorization and especially when considering diffusion models.

**Weaknesses:**

1. This work can be written more clearly, especially the section of the introduction was not very well written. I found that section 3.2 motivation was particularly helpful in setting the pace for this work.
2. In terms of the experimental setting, I do believe that performing experiments to see how the memorization ratio changes with repetitions in the data set might be a great way to further solidify if the method works. In particular, this could follow directly from the setup of Somepalli et al.
3. I would love to see more images of memorized inputs and further discussion beyond the two images shown in the paper right now to get a better sense of the performance of this method.
4. Most pertinently, I see that the mitigation strategies lead to a significant drop in CLIP score. In particular, if you were to look at the region on the plot between the model initialization before fine-tuning and the final fine-tuned model, it is evident that, especially when you contrast with Figure B where all the points are between 0.29 and 0.3, the inference time mitigation leads to a significant drop in CLIP score. It is unclear if this method is actually useful in that regard. It suggests that we are unable to reach the same performance as that of a model that was never fine-tuned. I am curious what the authors feel about this particular observation. In particular, a model that was not fine-tuned had a higher CLIP score on the prompts, but the method using mitigation achieved a much lower CLIP score. I am not able to position these results with the overall setup.

**Questions:**

See Weaknesses

---

> ### Author Response · Authors · 2023-11-21
> **Response to Reviewer 5xmi**
>
> We sincerely appreciate your valuable feedback and the time you've dedicated to providing it. Below, we address specific points you raised:
>
> > This work can be written more clearly, especially the section of the introduction was not very well written. I found that section 3.2 motivation was particularly helpful in setting the pace for this work.
>
> We appreciate your feedback on the clarity of our paper, especially the introduction. We will thoroughly revise these sections to enhance readability for the camera-ready version. Meanwhile, we are glad that you find the motivation section is helpful.
>
> > In terms of the experimental setting, I do believe that performing experiments to see how the memorization ratio changes with repetitions in the data set might be a great way to further solidify if the method works.
>
> We are grateful for your suggestion to investigate the memorization ratio with varying data repetitions. We have conducted additional experiments with different data duplication times: 50, 100, 150, 200, 250, and 300 times. The results with the SSCD similarity scores before and after applying the inference-time mitigation are presented in the table below. The proposed method is effective in all situtations with the same target loss for mitigation. We believe this is a good add-on for the future version.
>
> | duplication times |   50  |  100  |  150  |  200  |  250  |  300  |
> |:---------------:|:-----:|:-----:|:-----:|:-----:|:-----:|:-----:|
> |  No Mitigation  | 0.544 | 0.556 | 0.559 | 0.561 | 0.562 | 0.570 |
> | With Mitigation | 0.328 | 0.341 | 0.342 | 0.350 | 0.351 | 0.367 |
>
> > I would love to see more images of memorized inputs and further discussion beyond the two images shown in the paper right now to get a better sense of the performance of this method.
>
> We agree that more examples of memorized inputs would be beneficial. We have included additional memorized and non-memorized images and corresponding magnitudes of text-conditional noise prediction in Figure 8 and Figure 9 respectively in the appendix of the updated draft.
>
> > Most pertinently, I see that the mitigation strategies lead to a significant drop in CLIP score. In particular, if you were to look at the region on the plot between the model initialization before fine-tuning and the final fine-tuned model, it is evident that, especially when you contrast with Figure B where all the points are between 0.29 and 0.3, the inference time mitigation leads to a significant drop in CLIP score. It is unclear if this method is actually useful in that regard. It suggests that we are unable to reach the same performance as that of a model that was never fine-tuned. I am curious what the authors feel about this particular observation. In particular, a model that was not fine-tuned had a higher CLIP score on the prompts, but the method using mitigation achieved a much lower CLIP score. I am not able to position these results with the overall setup.
>
> We recognize your concerns regarding the observed decrease in CLIP score to the inference-time mitigation. This issue primarily arises from the model's significant overfitting to memorized data points, making it challenging to achieve high CLIP scores during inference without additional model retraining or fine-tuning. However, we want to emphasize that in practice, the model owner can deploy our proposed detection (section 3.3) and mitigation (section 4.2) strategy at the same time. Therefore, the mitigation will be only applied to memorized prompts, and, in non-memorization cases, the model performance will not be impaired.
>
> Meanwhile, we believe that a slight drop in CLIP score is still beneficial to the company when compared to the issues of copyright infringement or privacy violations due to outputting memorized training data.
>
> We hope our response can solve your concern regarding our paper. Please let us know if you have any more questions.

---

### Meta-Review · Area_Chair_xqfW · 2023-12-04

**Metareview:**

This paper addresses the critical issue of detecting and mitigating memorization in diffusion models, building upon prior work by Somepalli et al. and leveraging the impact of text prompts on model-generated images. Reviewers acknowledged the cleverness of the approach, emphasizing its speed and efficiency in detecting memorization without access to training data. The mitigation strategies proposed, both during inference and training, were deemed innovative, although concerns were raised regarding their impact on model performance and interpretability. Reviewers appreciated the rebuttal clarifications provided by the authors, resulting in updated scores from some reviewers. The paper's contributions in addressing memorization in diffusion models were acknowledged, but further improvements in various aspects were recommended for a more comprehensive understanding and application of the proposed methods. Some suggestions included improving clarity in writing, exploring the impact of mitigation strategies on model performance, and enhancing interpretability in detecting memorization. I recommend accept.

**Justification For Why Not Higher Score:**

N/A

**Justification For Why Not Lower Score:**

Based on the reviewers' comments and the positive aspects highlighted in their assessments, it's reasonable to recommend accepting the paper for oral presentation at the NeurIPS conference. The paper's contributions in detecting and mitigating memorization in diffusion models were well-received, showcasing innovative approaches and significant improvements over prior methods. Despite some minor concerns raised regarding clarity, the overall positive reception and valuable contributions warrant its candidacy for oral presentation.

---

### Decision · Program_Chairs · 2024-01-16

Accept (oral)